# Engaging Industry in Health Professions’ Education: Benefits and Challenges

**DOI:** 10.3390/ijerph20126131

**Published:** 2023-06-15

**Authors:** Belinda Kenny, David O’Connor, Ellie Sugden, Clarice Y. Tang, Caterina Tannous, Elizabeth Thyer

**Affiliations:** 1School of Health Sciences, Western Sydney University, Campbelltown Campus, Locked Bag 1797, Penrith, NSW 2751, Australia; 2Community Health Care, Primary Care Community Health, Nepean Blue Mountains Local Health District, Penrith, NSW 2750, Australia; 3Physiotherapy, Victoria University, Melbourne, VIC 3000, Australia

**Keywords:** health professions’ education, higher education, work-integrated learning, curricula

## Abstract

Effective partnerships between universities and industry facilitate health-profession students’ learning and work readiness. However, developing sustainable industry engagement in academic curricula remains challenging. This study utilised Social Exchange Theory (SET) to explore the benefits of and barriers to industry engagement within health-profession preparation programs. A realist evaluation framework was used to consider factors that impacted experiences and outcomes for academics and clinicians who engaged in the development and delivery of curriculum for a new health professional preparation program in speech pathology. A sequential mixed-methods design was adopted to explore factors influencing clinicians’ motivation to engage with the university, using an online survey (*n* = 18) and focus group (*n* = 5). Clinicians rated “personal development” and contributing to the “future workforce” as the highest personal benefits. “Sharing knowledge” was the highest team benefit, and “staff satisfaction” was the highest employer benefit. Time and workload were perceived barriers. Academics (*n* = 2) and clinicians (*n* = 3) who collaborated in learning and teaching experiences also participated in a post-engagement focus group. Three Context Mechanism Outcome configurations were shown to facilitate engagement outcomes: engagement as opportunity, partnership, and work readiness. In accordance with SET, the nature of exchange processes and professional relationships contributed to positive engagement outcomes for clinicians, academics, and enhanced health-profession education.

## 1. Introduction

The preparation of health professionals has long been considered a responsibility shared by both the educational institution and the profession. Health professions rely on the competent delivery of their services to maintain their reputation, status, autonomy, and public trust [1]. To achieve this, graduate competencies and attributes are determined in partnership with professional or regulatory bodies, and together they monitor and assure the quality of the educational program [2]. Often, this educational preparation of health professionals includes clinicians from the profession. In this article, we adopt the term “industry engagement”, using Manwaring et al.’s (2019, p.47) definition, as the relationship between universities and key employers and external stakeholders [3]. Learning how to effectively engage with industry may be pivotal to the development and implementation of a curriculum that prepares graduates for professional practice. Furthermore, engagement with academic programs may deliver benefits for industry partners. Surprisingly, there is limited evidence to guide academic–industry collaboration in health-profession education.

### 1.1. The Problem

There are many benefits of including clinicians in the educational preparation of health professionals, and recently this focus has shifted to include practitioners in the delivery of content at the university or in the development of the curriculum [4]. Clinicians may add a contemporary feel to a health-professional program. The literature suggests that their recent experiences in practice can help students understand the content and its application in service delivery and bridge the gap between theory and practice [5]. Contributions from practising health professionals in curriculum design, development, and implementation may foster innovation and ensure currency and responsiveness to changing healthcare needs and practices [6,7].

From a student-learning perspective, industry partners provide a window into the “real life” practice of applying knowledge and skills within dynamic health contexts [8], enhance active participation in reflective professional practice [9,10], and contribute to work readiness and employability [11]. As future employers of health science graduates, industry partners may enhance new graduates’ knowledge, skills, and professional values, enabling a ready transition from university to employment settings [12]. 

Practitioners’ insights ensure that health-professional graduates are prepared for the future workforce and realities of practice [13,14]. Thomas and Judd used a “community of practice” partnership between practice scholars and academic staff to review their occupational therapy curriculum and found that new graduates were potential “change agents with the profession” when their curriculum was developed in collaboration with practitioners [14] (p. 242). Garcia et al. also found that a collaborative approach to curriculum development was essential for the sustainability of their new occupational therapy program [15]. 

Clinicians may also benefit from engagement with health professions’ curricula. Workplace studies have demonstrated that opportunities to advance and pursue professional interests by engaging in diverse roles positively impacts health professionals’ job satisfaction and retention [16,17,18,19]. Opportunities for collaboration in developing and delivering a curriculum may address workplace challenges by providing clinicians with experiences that facilitate professional development and recognition of their practice-based knowledge. Knowledge sharing is also understood to not only benefit the individual but the immediate work group and broader organisation through reciprocity [20]. Such benefits may be highly valued by health professionals in contemporary practice who managed the significant workplace challenges that followed the global COVID-19 pandemic. Hence, further insights into how engagement in the scholarship of learning and teaching may leverage career development and satisfaction will be highly valued by industry partners and academics.

### 1.2. The Gap

Despite these positive benefits of industry engagement, there is a paucity of information on possible barriers or enablers to engagement and processes to develop sustainable partnerships that deliver quality learning and teaching experiences and meet the needs of industry partners [4]. While many health disciplines engage with industry partners to co-design or co-deliver elements of a program’s curriculum, such collaborations are typically built upon goodwill between individuals [21]. Evidence supporting industry engagement in health professions’ education has focused upon professional-placement learning experiences and research collaborations [10,22,23]. Whilst we acknowledge the essential role of work-integrated learning experiences, we argue a need for increased integration of workplace knowledge sharing and experience within the academic curriculum for health-profession students.

### 1.3. The Engagement Project

This study explored industry engagement in education through the development of a partnership underpinned by principles of Social Exchange Theory (SET) and provides the foundation for a long-term partnership, namely the Engaging Industry for Engaging Education project. Our study took an innovative approach to exploring industry engagement in the development and delivery of health professions’ curricula. Applying a SET framework, we explored concepts of “shared benefits” from academic and industry perspectives. We identified the perceptions of facilitators or barriers to engagement, with implications for developing sustainable partnerships between health science professional preparation programs and the local health industry.

The overall aims were to establish an innovative partnership to facilitate graduates’ work readiness, enrich students’ learning, provide professional development opportunities for practicing clinicians, and showcase the NBMLHD health service as an employer of choice for health-profession graduates. Industry and university partners concurred that a sustainable partnership will deliver longer-term benefits, including research collaborations, innovative learning, and teaching opportunities, and facilitate staff retention. The study addresses three research questions:What factors are rated by practicing speech pathologists as important facilitators and barriers to engagement in the academic curriculum of a professional preparation program?What is the nature of engagement experiences and outcomes perceived by industry and academic partners?How may industry and academic partners’ perceptions of engagement rewards and costs inform sustainable engagement?

## 2. Materials and Methods

A mixed-methods approach was used to facilitate an understanding of data collected at different stages of the research process [24]. The sequencing of quantitative–qualitative data assisted with data integration and enabled a purposeful investigation of engagement guided by a theoretical realist framework.

This study adopted a realist evaluation framework to consider “what works?” and “for whom, to what extent, in what context and how” [25,26]. Realist frameworks explore causal links between interventions and outcomes by developing context (C), mechanism (M), and outcome (O) configurations, or CMOcs. Context pertains to the “backstory” of the engagement and includes any social, cultural, historical, or geographical conditions impacting academic or industry partners. Mechanisms encompass strategies to facilitate engagement and responses to such strategies by both partners. Outcomes include intended and unintended consequences of these mechanisms [27].

### 2.1. Theoretical Framework

A realist evaluation is a theory-driven interpretative process that employs middle-range theories—in this case, Social Exchange Theory (SET)—to provide causative explanations for outcomes [28]. Social Exchange Theory (SET) is an influential conceptual paradigm for understanding workplace behaviour and relationships [29]. Although different views of SET exist, theorists agree that social exchange involves a series of interactions that are interdependent and generate obligations by both partners. Hence, reciprocity is a core element of continuing interactions.

A premise of SET is that perceived costs and benefits that accompany an individual’s interactions will determine how relationships are evaluated [30]. When rewards are perceived as relatively high compared with costs, engagement is perceived as positive and sustainable. However, if one or both partners perceive that costs increase or exceed benefits, then satisfaction and the motivation to maintain engagement wane. SET provides a lens for exploring factors that motivate or deter industry partners from engaging with a university. Two models of social exchange, proposed by Cropanzano and Mitchell, provide a helpful perspective for engagement between health and tertiary education partners [29].

Model 1 posits that the nature of the exchange process will shape developing partnerships: the process is grounded in reciprocity, which may build stronger interpersonal relationships between organisations, compared with processes where there are explicit duties, responsibilities, and tasks required by each partner.

Model 2 adopts an alternative perspective whereby relationships impact the nature of exchanges. As an interorganisational relationship evolves, the exchanges reflect developing levels of trust, commitment, and respect that guide what is given and received by each partner.

These two models are not mutually exclusive, and new partnerships may benefit from focussing on the nature of the exchange process and relationship building. In our study, the first SET model was applied to identify potential exchange interactions that are perceived to benefit university and industry partners. The second model aligned with the focus on collaborative planning, open discussion, and critical reflection by both partners throughout the study (refer to Figure 1). We drew upon contemporary frameworks for establishing effective university/industry partnerships which focus upon clear goals, expectations, and mutual respect [31,32,33].

In keeping with the realist evaluation, the study encompassed several stages: hypothesising about CMOcs, collecting data to identify motivating factors for engagement and explore engagement experiences, and analysing data to confirm or refute the explanatory Social Exchange Theory. We hypothesised that the advent of a new professional preparation program was an important contextual factor providing opportunities for developing an industry/academic partnership grounded in contemporary health-workforce needs. We acknowledged historical assumptions about the nature, value, and/or burdens of collaboration. The first part of our study aimed to understand exchange mechanisms from an industry perspective. We further hypothesised that managing the nature of proposed engagement activities and the engagement relationship were essential strategies to achieve positive engagement outcomes. By aligning proposed contributions with their professional experience, industry participants will engage in a meaningful and manageable manner and deliver quality student learning experiences. Engagement within a mutually respectful learning and teaching environment will then facilitate a positive, sustainable partnership. Expected outcomes were additional professional development experiences for participating industry partners and the development of authentic learning and teaching experiences for students. Furthermore, reciprocal engagement benefits will foster new engagement opportunities between industry and university partners.

Table 1 provides a study-specific definition of contextual factors, mechanisms, and outcomes relevant to this study.

The Western Sydney University Bachelor of Speech Pathology (B. SP.) four-year undergraduate program commenced in 2020, and the Engaging Industry for Engaging Education (Engagement Project) commenced with second-year content offered for the first time during 2021. Engagement focussed on three discipline subjects addressing the assessment and management of children and adults with diverse communication and mealtime needs, including lectures (theoretical), tutorials (applied clinical reasoning), and practical (skills-based) classes. In response to co-occurring health restrictions, the curriculum was delivered both face-to-face and online. All clinicians opted to collaborate with the design and implementation of tutorial, case-based reasoning and/or practical components of the curriculum that closely aligned with their professional experience. Additionally, clinicians engaged in discussions regarding the theoretical content presented in lectures and shared topic-relevant practical resources.

### 2.2. Participants and Recruitment

In this study, we defined the CMO “case” as practicing speech pathologists employed within a large (~9000 km^2^) state-funded health district covering urban and semi-rural areas, servicing ~350,000 people, including hospital and community health contexts and the associated university academics. For clarity, in the manuscript, industry staff members are referred to as clinicians, and university staff members are referred to as academics. The sampled case population had a workforce of 32 full- or part-time speech pathologists (clinicians) and 3 speech pathology academics (academics).

As Figure 1 indicates, there were three stages of the study, comprising a survey, the engagement phase, and a focus group. Two participant groups were recruited to the study. Recruitment for the survey was via staff email lists to all speech-pathology clinicians employed within NBMLHD and included on the speech-pathology staff list. Recruitment for the post-engagement focus group sought to include B.SP. academics and clinicians actively engaged in curriculum co-design or co-delivery during 2021, and, as such, we employed a purposive sampling method with email contact made with participants via an independent member of the research team.

This study complied with the Australian Code for the Responsible Conduct of Research (2018). The code articulates the broad principles and values that characterise a research culture, including respect for others, research merit, integrity, justice, and beneficence [34]. 

### 2.3. Research Tools

The Engagement Survey tool was developed by the authors to explore clinicians’ perceptions of potential benefits (rewards) and barriers (costs) of engagement. An iterative design process was used to reflect interdisciplinary, academic, and industry engagement perspectives and enhance readability and face validity. A purpose-designed survey was required, as no validated scale was available to collect the necessary data.

The online survey comprised 22 questions that addressed potential individual, employer, and workplace benefits and barriers to engagement. Respondents rated statements on 5-point Likert-type scales that ranged from “not important” (1) to “extremely important” (5) for benefits and from “strongly disagree” (1) to “strongly agree” (5) for barriers to engagement and factors supporting sustainable engagement practices (Appendix A). Survey-question content was derived from industry engagement, tertiary education, and the work-integrated learning literature [11,17,31]. Open-ended questions enabled participants to identify perceived benefits or barriers not captured by survey questions.

The semi-structured online engagement focus group explored clinicians’ and academics’ perspectives of engagement experiences and outcomes. A secure online platform (Zoom™) ensured flexibility for focus-group participants who were employed across different sites. The focus group was facilitated by an independent research assistant who was experienced in qualitative interviewing and realist methods. Relevant works from the literature informed topic questions that addressed expectations and experiences with curriculum engagement. Perceived outcomes and recommendations for sustainable engagement were interrogated during the session. The facilitator used a collaborative form of theory refinement that focused on participants’ experiences and strategies for developing mutually beneficial, sustainable partnerships [35].

### 2.4. Data Collection and Analysiss

The Engagement Survey was hosted on the Qualtrics XM survey platform (Qualtrics, Provo, UT, USA 2020). Survey data were downloaded to Microsoft Excel^®^ and entered into IBM SPSS version 26 (Chicago, IL, USA). Nominal and ordinal data were coded for statistical analysis. Due to the low frequency of open-ended responses, these data were not subjected to qualitative analysis but used descriptively to complement survey findings.

The engagement focus group was audio recorded and transcribed verbatim by the independent research assistant. Focus-group data were coded by two authors (BK and ES) and analysed using a realist CMOc framework that was consistent with the methods described by Gilmore and colleagues [36]:

Context—individual participant factors that may positively or negatively influence engagement experience;

Mechanisms—collaborative strategies, learning and teaching approaches, and resources implemented during subjects to facilitate engagement and participants’ responses to these approaches;

Outcomes—causal patterns identified when different contexts and/or mechanisms experienced by participants were associated with particular engagement outcomes.

The two aforementioned authors jointly coded approximately 20% of the transcription data to develop coding rules. Then, they independently coded the remaining transcript and met to compare and discuss codes. Differences in coding were resolved by returning to the transcript and via consensus. A decision-making audit trail facilitated consistency in coding. Following coding, the authors examined findings against hypothesised CMOcs. The analysis was presented, and the CMOcs were further refined via discussion with the wider research team.

### 2.5. Rigour

The analysis was an iterative process to test, refine, and verify our CMOcs [26]. Throughout the data collection and analysis, adherence to the TAUPAS (Transparency, Accuracy, Purposivity, Utility, Propriety, Accessibility, and Specificity) criteria, including the use of the RAMESES II reporting standards, was maintained [37,38].

## 3. Results

The findings from this study identify factors perceived as important facilitators and barriers to engagement in academic curricula by clinicians and provide potential pathways to sustainable industry engagement in the professional preparation of health professions graduates.

### 3.1. Engagement Survey

Clinicians’ perceptions of the potential benefits and barriers to engagement were quantitatively analysed via an anonymous online survey. A total of 21 clinicians (66% of eligible employees) responded to the Engagement Survey. In all, three respondents answered the demographic responses only (excluded from analysis), and the remaining 18 participants (56%) completed all sections of the survey. Participants’ responses were included if they completed all ratings. Three participants had missing data for one or two demographic data items (see Table 2).

Table 2 shows clinicians’ demographics. When asked to indicate their awareness of the Engagement Project, ten (56%) respondents stated that they were aware of the project’s aims. In response to potential interest in engagement with the university, seven respondents (39%) expressed interest in immediate engagement. A further ten respondents (56%) expressed an interest in future engagement. Overall, 95% of respondents expressed potential interest in university engagement.

### 3.2. Perceptions of Benefits

#### 3.2.1. Individual Benefits

Figure 2 presents the perceived individual benefits of engagement among the 18 participants.

Overall, all except one participant perceived all survey benefits as somewhat important engagement factors. *Development of future workforce* (median = 4, IQR = 0.5) and *personal development* (median = 4, IQR = 1.0) were the highest rated individual benefits for clinicians, with more than half the total participants rating these factors as “extremely important” benefits. Open-ended responses accorded with survey ratings, with inspiring others to join the workforce, supporting student placements, and improving connections between the university and local health district described as engagement benefits.

#### 3.2.2. Team/Organisational Benefits

Regarding perceived benefits to the team and organisation, participants’ ratings indicated all listed factors to be somewhat important (Figure 3). *Two-way sharing of knowledge* (median = 5, IQR = 1), *boosting job satisfaction among existing staff* (median = 5, IQR = 1), and *developing new graduate work readiness* (median = 5, IQR = 1) received the highest mean ratings as benefits for the team and organisation.

Clinicians’ responses to open-ended questions offered similar perceptions of team and organisational benefits. Further development in education and research was raised as a valued benefit. Moreover, career-related benefits, including workplace diversity, achievement of performance goals, and strengthening career pathways, were considered potential team benefits.

### 3.3. Perceptions of Barriers

#### 3.3.1. Individual Barriers

Participants rated their agreement/disagreement of specific factors as individual barriers to engagement. As Figure 4 illustrates, *time commitment* (median = 5, IQR = 1) and *management of clinical workload* (median = 4.5, IQR = 1.75) were identified as barriers to engagement. Participants appeared to be somewhat confident about their ability to utilise online educational technology (median = 3, IQR = 1) and interact with other university academics (median = 3, IQR = 1), as most participants neither agreed nor disagreed with these two factors.

#### 3.3.2. Team/Organisational Barriers

Workplace barriers reflected concerns that participation in university activities may increase colleagues’ workload, with over 80% of respondents “agreeing” or “strongly agreeing” that engagement may negatively *impact other’s workload* (Figure 5). *Inadequate workplace support* (median = 3.5, IQR = 2) and *reduced individual productivity* (median = 3, IQR = 2) were other identified barriers. However, overall, engagement was widely perceived as being valued by team members and managers, with only one participant “strongly agreeing” that university engagement will not be valued by managers.

The participants did not provide any additional information regarding individual or workplace barriers in response to open-ended questions.

### 3.4. Perceptions of Factors Facilitating Sustained Collaboration

We were interested in clinicians’ perceptions of factors that underpin sustained and effective collaboration between university and industry partners. Figure 6 shows that the participants rated diverse factors that are important for supporting engagement. *Opportunities for knowledge and information sharing* (median = 5, IQR = 0.8), *mutual benefits for clinicians and academics* (median = 5, IQR = 0.8), and *clear expectations for engagement* (median = 5, IQR = 2) were key factors for collaborative industry/university relationships.

When asked about activities to motivate future engagement, 14 out of 18 (78%) participants expressed willingness to be involved in clinical education with students. Furthermore, participants reported interest in contributing clinical scenarios for student learning or assessment (*n* = 11, 61%). They reported relatively less interest in the delivery of professional podcasts and designing an academic curriculum for topics related to their professional practice, with only four participants (23%) expressing interest in these educational activities.

## 4. Engagement Focus Group

To develop deeper insights into factors that facilitate sustainable industry engagement in health professions’ education, we explored the experiences of academics and clinicians to determine the contexts and mechanisms that led to their engagement outcomes. The focus group comprised the three clinicians and two academics who engaged in curriculum development and delivery during the time of the study. The clinicians were experienced professionals who were employed full time; two from were from a hospital setting, and one was from a community health setting. The two academics were employed in full-time continuing positions within the university and were experienced in curriculum design and learning and teaching in face-to-face and online contexts. All of the focus-group participants were women. None of the participants had directly collaborated prior to the study.

In accordance with our realist approach, contexts, mechanisms, and outcomes were configured into causal CMO configurations. We identified three CMO configurations that reflected engagement experiences. The first CMOc focuses on industry engagement as opportunity for clinicians and academics. The second CMOc focuses on relationship building for a collaborative partnership. The third CMOc addresses learning aspects of engagement, with a focus on work readiness. Figure 7 presents three configurations that we propose will facilitate sustained industry engagement in health professions’ education, along with supporting evidence from focus-group codes. The following section describes each CMOc with deidentified exemplars from the clinicians (coded C1, C2, and C3) and academics (coded A1 and A2) who participated in the study.

### 4.1. CMO1 Engagement as Opportunity

CMOc1 framed participation in the Engagement Project as an opportunity for professional growth and development.

“A really good opportunity to share what I’ve learned at work in a hospital, and I haven’t really had much to do with students, and university other than when they’ve come to placement, so it was kind of nice to do more on the theory” (C1).

Context: Clinicians expressed motivation to engage in new professional challenges. They sought to extended clinical education roles by working in new ways with students in academic settings: “a new experience under my belt” (C1). Furthermore, clinicians perceived university engagement as providing opportunities to share professional knowledge and skills beyond their workplaces. For academics, engagement provided opportunities to make connections with local industry partners and, similarly, interact beyond the realm of clinical placements:

“We often think about clinical placements, but I really wanted to think about all the different ways that we could potentially engage with our industry partners here within the learning and teaching space” (A2).

However, clinicians identified caseload demands and competing priorities as potential barriers to translating motivation to active contributions in the Engagement Project.

Mechanism: Harnessing clinicians’ motivation required academics to offer new professional experiences as engagement options. For example, clinicians identified working on theoretical concepts relevant to their practice and shaping a curriculum as valued professional experiences. However, participants suggested that individual clinicians need support to manage personal and professional factors limiting engagement.

“There’s a big leap from working in practice to entering the world of academics. And you know, realising that we have a lot of clinical experience, so a lot that we can share in that sphere, but then maybe overcoming some of those initial barriers and hurdles around what stops us moving on to exploring some of those opportunities” (C2).

Promoting involvement with other team members was proposed as a strategy to overcome such barriers.

“Sharing how it’s been a positive experience and promoting it within our teams will help sustain clinicians’ interest and excitement about being involved” (C3). As this clinician explained, “human nature, we tend to view something new as being extra work or potentially burden or hard but if the people who’ve done it before you can share how I found it valuable and rewarding then you’re more likely to want to do it [too]”.

Outcome: Clinicians and academics described engagement as a strategy to inspire and enliven their current professional roles. Engagement provided two positive outcomes. The first was new learning, with clinicians developing additional education skills and academics acquiring new clinical insights. Secondly, engagement with colleagues from another sector was psychosocially rewarding and provided an invigorating element to work practice.

“I personally felt like I had something to offer, and that it was well received and reminded me that what we’re doing now every day is of interest to others, so that did help you feel valued” (C2).

Importantly, both partners suggested that a sustainable partnership required ongoing exploration of engagement opportunities. Clinicians recommended building networks within the university, accessing library resources, and collaborating in research.

“One of the potential things that we could do is maybe feel more engaged with the university whether that’s around evidence-based practice, writing CATS [critically appraised topics] … or having connections with people in that setting” (C2).

Academics considered different areas of curriculum design and implementation that may benefit from industry input and potential research collaboration.

### 4.2. CMO2 Engagement as a Partnership

CMOc2 reflected an emphasis on the engagement relationship. Participants reiterated a need for collaborative interactions to facilitate the engagement experience.

“It didn’t feel like you were just coming in and that the lecturer disappeared, but it did feel like it was a joint process and that there was facilitation happening” (C2).

Context: Clinicians and academics acknowledged that a new academic/industry partnership requires learning about each other’s roles and the mutual benefits of closer ties between academics and clinicians.

“We have a lot of overlaps, but also a lot of differences as well, so I think just knowing what roles we have and how we can work together” (C1).

Academics concurred that it is partly getting to know each other and becoming familiar with each other in terms of developing professional relationships.

“Because that takes time, understanding what it is that we can both offer” (A2).

Participants perceived that they offered complementary skills and experiences necessary to build a collaborative partnership. Therefore, learning how to work together incorporated understanding ways of sharing knowledge and experiences and being responsive to workplace constraints.

Mechanism: Clear and reasonable expectations were perceived as integral to effective collaboration. Clinicians explained the importance of setting reasonable expectations and time frames.

“Having really clear expectations of what is reasonable and practical to provide from a clinician’s point of view and lots of time to prepare for that, because sometimes time can be a barrier for us when we have a lot of competing demands, so I really appreciated having a lot of time to plan and prepare for my contributions” (C3).

Furthermore, clinicians sought meaningful and efficient contributions.

“The benefit of having clear expectations is not doubling up on things, but also knowing how we can maximize what we can offer the students from a practical point of view” (C1).

Academics shared the importance of open and transparent expectations:

“Really clear expectations help with everyone’s planning, and it also helps [the engagement become] more manageable and kind of less daunting and overwhelming” (A1).

Therefore, both partners perceived that effective communication underpinned positive and potentially sustainable engagement.

“If we’re communicating with each other, we’ll understand what each other’s needs are and what’s reasonable and achievable for each of us to supply”. Academics described a need to navigate (A2) curriculum structure and clinician workflow demands to identify the optimal timing for engagement activities. C2 affirmed that different options enabled her to select the type and timing of engagement that was going to best fit her capacity.

Clinicians and academics consistently identified feedback as a core engagement tool. They adapted the nature of feedback provided to colleagues within an academic context. One academic noted, “giving feedback to a colleague is very different, [to providing] feedback to students, which is how I normally [work]” (A1). Clinicians also reflected upon feedback processes with academic colleagues:

“How do we receive feedback and how do we ask for feedback? Which is from an adult learning perspective, a useful process for us to think about” (C2).

Clinicians reported an openness to and appreciation for feedback from academics based upon professional trust.

“The relationship that the lecturer I’m sure took great pains to establish was very respectful, mutually respectful so I felt that any feedback she gave me I was willing to hear and willing to action” (C3).

Importantly, clinicians perceived feedback as a bidirectional process.

“I got to give [academic] some input and some resources that she could use in her lecture so I thought that was really good that we could both give feedback to each other” (C1). A1 concurred that receiving clinicians’ feedback on the curriculum was really valuable.

Feedback reportedly facilitated clinicians’ confidence to transfer their educational skills from the workplace to the academic environment. Furthermore, post-engagement feedback confirmed that their contributions were valued by academics.

“Feedback at the end of all that process, made it seem like it was really worthwhile and really what we did was quite meaningful” (C1).

“Thus, feedback becomes an important part of the process as well to feeling like it is a partnership and, I certainly felt valued for being involved in that process… getting that feedback as an adult engaging in any process of learning and where you hear some feedback about how you’re going, that is important” (C2).

One academic (A2) suggested that feedback at an organisational level may help sustain engagement by garnering support for participating clinicians.

Outcome: Understanding that new collaborations involve *trial and error* (C3), establishing clear expectations, and applying appropriate feedback processes was perceived as an essential foundation for a respectful and responsive partnership.

“Forming the relationships is the foremost thing that comes to mind as a benefit for everyone. Jumping that hurdle between academia and research and practice” (C2).

This clinician explained that having just one link in this sphere of the academic world allows you to” then feel that you have a connection to ask a question or say who the best person is to direct this query or this interest to”, taking a step toward sustainable partnership.

### 4.3. CMO3 Engagement for Work Readiness

CMOc 3 focused on student learning as a driver for engagement. Academics perceived that student learning is enriched by practicing health professionals.

“I really wanted for our students to… learn from a broader range of people who have a much broader range of experiences than I have” (A1).

Enhancing student learning was a shared aim.

“I wanted to be able to provide an experience that is valuable and practical and helpful” (C3).

Context: An important contextual feature of this project was that many students enrolled in the B. SP. resided within the participating LHD and, as such, will potentially attend professional placements and/or seek future employment with the industry partner. However, at the time of the study, students had not commenced external professional practice placements, and one academic (A1) identified students’ limited professional experience as a factor in planning engagement activities:

“That was a bit of a challenge for our students’ learning to realise that people can be more complex and that you need to have expertise in a lot of different areas and skills to be able to manage clients”.

Essentially, students were progressing on their journey from novice clinicians to competent future health science graduates. Academics needed to prepare students to manage the cognitive load of applying knowledge and skills in scenarios that include professional complexity during early stages of their program. Clinicians reported similar tensions with presenting real-life practice at the right level for optimal student learning.

“What might be in my mind feeling like is not a complex client is actually still a very complex client for someone … in second year. The goal being not to overwhelm students, but instead to share and motivate and inspire” (C2).

Experienced clinicians acknowledged that significant time had elapsed since their own university education.

“Tailoring what we know to be in a tutorial form [or] in the form of assessment task. It’s been a while, since I’ve been at university so having to bring myself back to that and remember… where students are up to and how much to pitch to them” (C1).

Both partners agreed that students’ learning experiences must bring the professional workplace into the classroom.

“Really authentic for our students and really help to prepare them for real life practice. And by that, I mean to succeed in their clinical placements but also thinking about these students as future new graduates” (A2).

Mechanism: Preparation was perceived as a key strategy for the successful engagement of clinicians in students’ academic learning.

“We need to have clear communication and making sure that we are understanding what previous lectures, what topics have been covered” (C2).

Academics suggested that greater access to online learning and teaching resources may ensure that academics, clinicians, and students are “all on the same page” (A1). Furthermore, academics noted the importance of preparing students for industry engagement in their learning by explaining that this was a unique learning opportunity. For example, A1 “spent a lot of time with the students before the guests…to talk about how important it was … to get some other perspective [and to facilitate] students’ engagement and excitement for industry-led learning experiences”.

Academics reflected on the benefits of attending class during clinicians’ teaching.

“Really understanding the experience, the students were having because they were excited about it afterwards and they even had some extra questions and some comments after the presentation and having been there and part of it, it was much easier to respond” (A2).

Clinicians echoed the importance of working together in the same learning space to manage the continuity of student learning between academic and workplace settings.

“It was valuable to have [academic] there to not only just legitimize what I was talking about by either supporting it or drawing parallels to thingsthat she’d experienced” (C3) but to discuss and refine learning activities for the next group of students. A process of knowledge exchange between partners was perceived as a key strategy for developing authentic student learning experiences.

“It’s the benefit of the knowledge exchanges that we also learn how to teach our students, how to apply and integrate theory with the practical things that we know in the hospital, that we do very well in” (C1).

Preparation included navigation of the technological aspects of engagement. In our study, an unanticipated challenge was that the online learning platform utilised by the university was not readily accessible to clinicians due to health-department firewalls.

The development of sustainable resources was a recommended strategy for efficient engagement and to support clinicians who may be interested in future participation.

“Especially for new people who might be new to this they might be asking what is it that I’m expected to do? What’s been done before? It’s often helpful to see what other people have done to give you a framework to work from” (C3).

Outcomes: Clinicians perceived diverse positive outcomes from engagement, with multilayered student-learning benefits. Crystallising students’ understanding of the benefits of their chosen profession was at the forefront.

“It is based on real life which hopefully inspires and motivates students to understand a bit more of the role of a speech pathologist” (C2).

The academics agreed:

“Helps to maintain their passion for the profession…Halfway through the program you know, keep going, this is what I enrolled for, this is what I’m really passionate about” (A2).

Clinicians reflected on their willingness to provide contributions they would have valued during their own professional preparation:

“When I was a student, I would have liked to have guest lecturers talk to me about their experiences so … being able to hear from someone who was working there, I think that would have been really helpful for me as a student” (C1).

There were further positive outcomes associated with knowledge exchanges between academics and clinicians. Clinicians perceived that their contemporary workplace experiences enabled them to offer enriching professional practice scenarios that contributed to students’ holistic skill development.

“An appreciation of the complexity of some of the things that we do clinically that might not come through when you’re looking at theoretical information or evidence-based practice approaches to treatment. So, it’s sort of the psychosocial aspects of what we do as speech pathologists [as] another benefit for the knowledge exchanges” (C3).

Correspondingly, clinicians perceived that the educational skills they acquired in academic settings could facilitate their skills as clinical educators in their work settings.

“As clinical educators of students on placements we’re not necessarily trained as adult educators, so we can benefit from exchanging knowledge with the university because that’s their area of expertise” (C3).

From academics’ perspectives, engagement enhanced the quality of students’ education and provided insights into their profession:

“Getting them ready for their placements, getting ready to be graduates and qualified. Learning from how people just work in different places is really valuable” (A1). “Moreover, engaging with industry may inspire students to follow clinicians’ career path broadening their awareness of different [career] options” (A1).

None of the participants expressed negative outcomes from engagement.

“It’s been a good initiative and hopefully it will keep going” (C2).

## 5. Discussion

This study evaluated a collaboration between speech pathology academics and clinicians from NBMLHD to develop a mutually beneficial partnership that supports professional preparation and readiness for healthcare practice. Social Exchange Theory was applied to hypothesise and explore factors that influenced engagement experiences and outcomes.

The survey findings demonstrated strong motivation for practising clinicians to engage with academic colleagues in a professional preparation program. SET posits that individuals may assign different values to expected rewards. In our study, the perceived benefits focused on individual career aspirations and/or development of the profession. Clinicians’ identified individual professional benefits aligned with Boocock and O’Rourke’s findings that workplace role diversity impacts career development and job satisfaction [16]. Importantly, contributing to the development of the future health workforce was perceived as a personally fulfilling benefit of engagement. SET accepts that an individual’s actions are not solely driven by self-interest, and accordingly, the clinicians in this study were motivated to “give back” to professional preparation by contributing learning experiences they would have valued as students. Participants perceived that academic and industry engagement may bridge gaps between theory and practice and facilitate students’ emerging professional identities.

Individual barriers, including time commitment and workload management, were unsurprising given the nature of clinicians’ busy caseloads. Workload and caseload size have been previously identified as factors impacting health professionals’ job satisfaction [39]. Nonetheless, the focus-group participants suggested that individuals must reflect on quasi-barriers to engagement, including time, that may be readily overcome by effective communication and planning. Addressing and changing the perceptions of barriers may help individuals avoid letting collaborative projects remain within the purview of a small team of initiating academics and clinicians. 

Bidirectional giving and receiving is inherent in SET, and respondents focused on the two-way sharing of knowledge, skills, and resources between industry and university partners. Our findings showed that clinicians are highly motivated to adapt their clinical education skills to academic settings and extend their adult learning skills through collaboration with academic partners. While the importance of clinical education roles cannot be underestimated, clinicians were clearly interested in diverse engagement options with theoretical, evidence-based interactions. Framing engagement as an opportunity (CMOc1) rather than a burden may motivate clinicians to engage in new ways with a university. Continuing professional development is a core aspect of health professional practice [40], and the academics and clinicians reportedly valued sharing knowledge and experiences as a participatory form of intellectual inquiry. Furthermore, our findings suggest that external recognition may be an important engagement incentive for clinicians who typically work in demanding clinical environments. Recognition and acknowledgement are factors influencing health professionals’ well-being, job satisfaction, and intention to stay in or leave their professions [18,39].

In keeping with a SET framework, strategies that emphasise rewards and address engagement costs may influence clinicians’ perceptions of the relative value of rejuvenating professional experiences compared with time demands. Increased autonomy over work processes and experiences may facilitate job satisfaction [39]. Moreover, in their study of factors impacting perceptions of embeddedness in the speech pathology profession, Heritage, Quail, and Cocks [40] found that opportunities for career advancement and the development of professional networks were important factors determining intention to stay in a profession. Hence, participation in innovative workplace opportunities may help retain team members who seek professional challenges and role diversity. Positive correlations between job satisfaction and workplace performance may therefore deliver benefits to teams and organisations that outweigh time commitments.

An underlying expectation of SET is that successful initial experiences will predicate sustainable engagement. However, our findings suggest that, for engagement to be perceived as an opportunity, new and different options must be available to offer continued quality professional development opportunities for clinicians and academics. Clinicians who opted to collaborate engaged in the design and delivery of tutorial workshops and practical classes. Participation in podcasts, focussed interviews, or discussion panels was neglected, and perhaps academics may need to focus more on presenting these experiences as interesting and achievable professional development opportunities. Including both low-level engagement (indirect and/or limited contact, including guest lectures and simulation assessments) and high-level engagement options (external advisory committee and co-design of curriculum or delivering curriculum modules) will be a key strategy for engaging clinicians and academics, from early career to senior professional experience groups [3]. Our findings resonated with previous studies that demonstrated an appetite for scholarly discussion and research amongst allied health professionals [14,41,42]. Building stronger links between learning, teaching, and research engagement options will be explored during future iterations of this collaborative project. The development of a curriculum for a research engagement pathway may have significant long-term benefits for both partners. 

The contemporary applications of SET focus on relational rather than purely transactional exchanges between individuals [29]. The engagement as partnership CMO2 reflected the importance of a mutually respectful relationship that acknowledged and responded flexibly to the capacity of individuals and their workplace demands. Feedback was perceived to be an essential process, strengthening the developing relationship. Given that feedback was highly valued as a reciprocal exchange tool and its role as a motivating factor for adult learners [43], further attention to the nature and timing of feedback will be incorporated into future engagement activities.

Enhancing student learning experiences was a key focus of this study (CMO3). The clinicians were experienced clinical educators, but they had no prior experience with university teaching and reported a significant time frame since their own university experiences. They openly discussed the challenges of determining levels of complexity from a student perspective and concerns regarding overwhelming rather than inspiring students. Collaborative educational approaches require a measure of tenacity. Future engagement may benefit from open discussion regarding any underlying anxieties when planning experiences outside clinicians’ and academics’ typical educational practice. 

Our findings align with Cropanzano and Mitchell’s interlocking models of social relationships. A reciprocal exchange process created professional and interpersonal bonds between participants, as reflected in descriptions of engagement experiences as “inspiring” and “enjoyable” [29]. A shared commitment to delivering quality professional preparation learning experiences within a safe adult learning environment may then evolve the nature of future exchanges. After a successful outcome, participants may more readily try engagement experiences they perceive to be more challenging when they feel supported within a trusting, collaborative partnership.

A benefit of realist evaluation is that findings can enable or change existing reasoning about an innovation [44]. Project findings may change reasoning when participating clinicians and academics adopt the role of workplace champions and discuss the positive outcomes they experienced through engagement. Colleagues who are ambivalent regarding the project’s benefits may acquire an understanding of the value and importance of engagement in relation to their career development. 

## 6. Limitations

This study presents a foundational step in a long-term engagement project between industry and academia. The survey findings reflect the perceptions of the current LHD speech pathology workforce regarding their perceptions, interest, and expectations for engagement with the university. The survey did not capture the views of new graduates, academics, or colleagues from other health disciplines.

Focus-group data were gathered from the three clinicians and two academics who participated in the first iteration of the engagement project. This is an important contextual factor, and further research is needed to determine the influence of engagement factors on sustainable social exchanges and the longer-term outcomes of collaboration.

An absence of data on student learning outcomes is a limitation of this study. Student feedback suggested that learning experiences provided insights into professional practice and case studies as authentic learning experiences. Further research may show if engagement experiences engender students’ passion for the profession and present the LHD as a preferred employment option.

### 6.1. Engagement Implications

Our findings offer some SET-based guidelines for initiating engagement between universities and industry partners in academic settings by considering factors that may affect the nature of the exchange and the quality of the professional relationship between partners.

### 6.2. Exchange-Process Factors

Develop a range of engagement options.Provide different timeframes for engagement.Allow adequate preparation time.Include learning experiences that provide an opportunity to explore complexities of health professional roles and responsibilities beyonda “guest lecture topic”.Focus on continuity between academic and workplace learning so that industry contribution is integrated with the curriculum and not perceived by students as an “add-on”.

### 6.3. Exchange-Relationship Factors

Leadership from both partners overtly values engagement;Clear articulation of perceived benefits and barriers to effective engagement;Openness to trialling new ways of working together;Reciprocal feedback and reflection during and after engagement;Mutual recognition and acknowledgment of each partner’s educational skills and experiences.

### 6.4. Future Directions

The Engaging Industry for Engaging Education project is continuing to grow, with 11 clinicians and 4 academics planning or currently involved in a range of collaborative assessment and learning and teaching activities. Early signs of sustainability are characterised by clinicians’ involvement in simulation, preparation of students for placement, and development of future curriculum focusing on employability and work readiness. A research partnership with co-supervision of a student from the program’s inaugural honours program has initiated a practice-led research pathway between industry and academic partners.

From the Lead Clinician’s perspective, future developments will attract staff members who represent different services, expertise, and levels of experience within the LHD to engage with the university. Both industry and academic partners wish to facilitate the uptake of diverse educational and professional experiences. The academic partner is motivated to pursue opportunities to develop research collaborations.

Further studies may explore how the three CMO pathways may provide a framework to facilitate industry engagement in health professions’ education in other settings.

## 7. Conclusions

In a realist evaluation, researchers hypothesise the mechanisms that are likely to operate, the contexts in which they operate, and the expected outcomes. During our project, we hypothesised that a new professional preparation program would provide a unique opportunity for industry engagement. We expected that providing engagement options, flexibility, and strategies to enhance communication would initiate and maintain engagement between academics and clinicians. By employing such strategies, we intended to provide mutual benefits for both partners. Combined findings from the survey and focus group confirmed our hypotheses and provided deeper insights into what worked for whom, how, and in what contexts.

SET provided a lens for exploring potential benefits and barriers to engagement. Understanding the perceived value that clinicians assigned to potential benefits and perceived importance of barriers to engagement can facilitate the ongoing development of mechanisms that shift the reward/cost ratio of collaboration. Importantly, clinicians and academics realised positive outcomes, and this study showed that there are many ways in which industry and academic partners may engage.

Social systems must be conceptualised as open, not static, systems. Individuals, perspectives, information, and resources will flow in and out of a system, and therefore simple, linear approaches to evaluation may fail to capture the complex causes that lead to outcomes [35,44]. In this study, positive outcomes were underpinned by the nature of opportunities afforded through the project, developing strong professional relationships and a shared goal of enhancing the professional preparation for future health-profession graduates.

Realist evaluation is an iterative process, and, as such, repeating the process of engagement in different settings with a different cohort of academics and students will facilitate affirmation or contradiction of the CMOcs in our findings.

## Figures and Tables

**Figure 1 ijerph-20-06131-f001:**
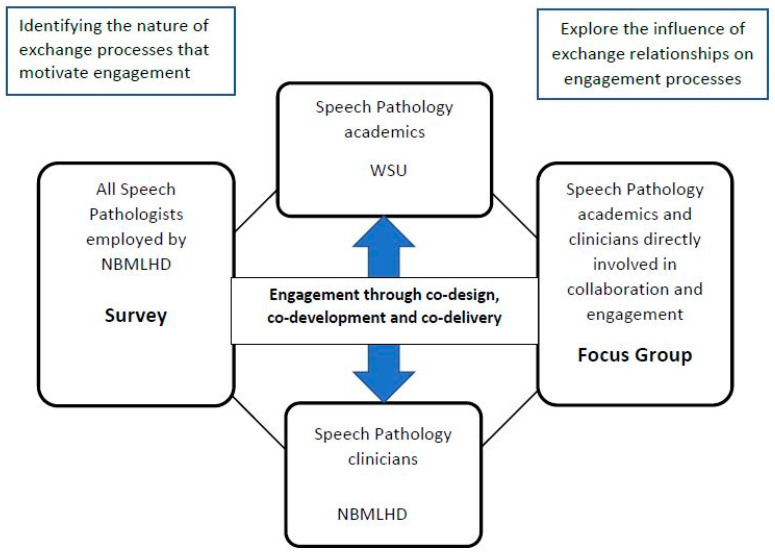
Participant involvement in three stages of the study.

**Figure 2 ijerph-20-06131-f002:**
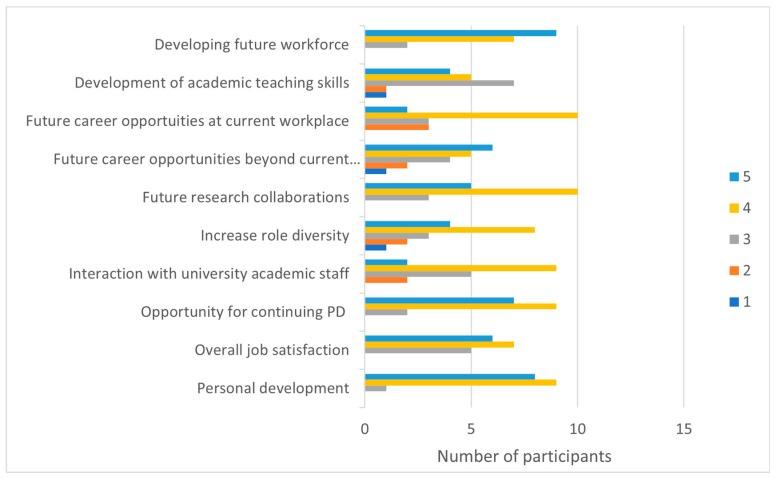
Engagement Survey: Perceived Individual benefits. Rating: 1 = not important, 2 = some importance, 3 = somewhat important, 4 = very important, and 5 = extremely important.

**Figure 3 ijerph-20-06131-f003:**
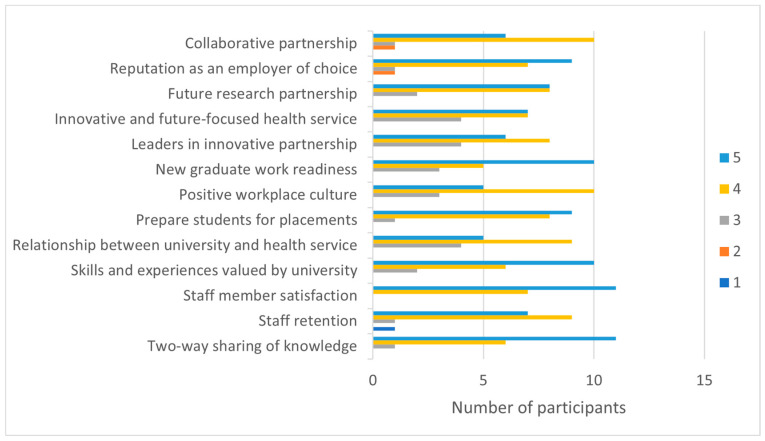
Engagement Survey: Perceived benefits for team and organisation. Rating: 1 = not important, 2 = some importance, 3 = somewhat important, 4 = very important, and 5 = extremely important.

**Figure 4 ijerph-20-06131-f004:**
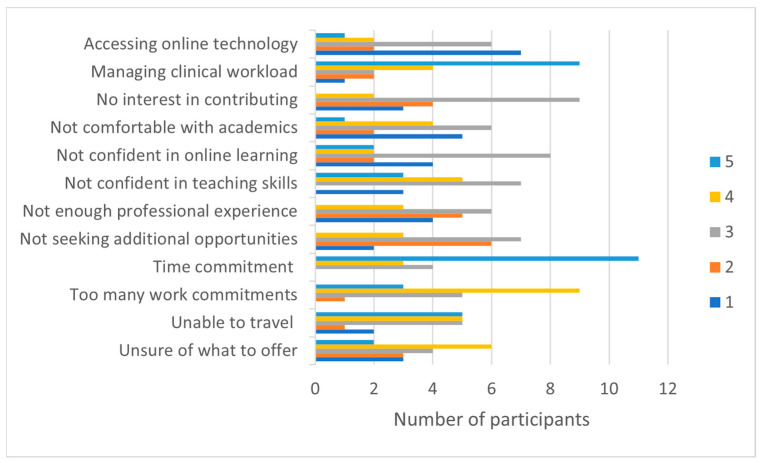
Engagement Survey: Perceived individual barriers. 1 = strongly disagree, 2 = disagree, 3 = neither disagree nor agree, 4 = agree, 5 = strongly agree.

**Figure 5 ijerph-20-06131-f005:**
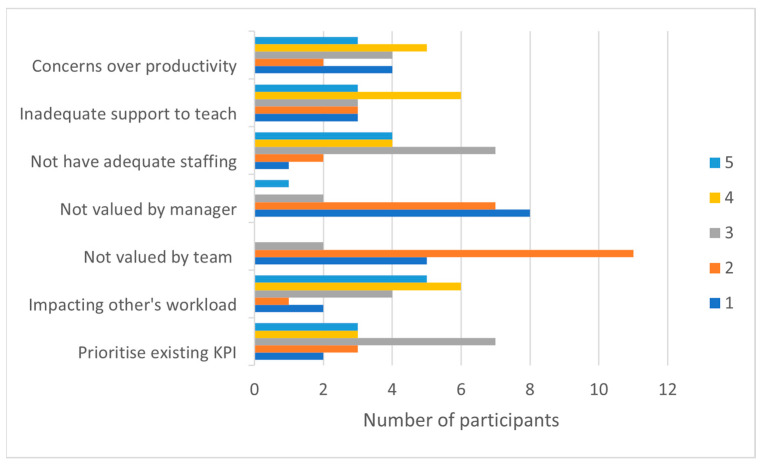
Engagement Survey: Workplace barriers to participation. Rating: 1 = strongly disagree, 2 = disagree, 3 = neither disagree nor agree, 4 = agree, and 5 = strongly agree.

**Figure 6 ijerph-20-06131-f006:**
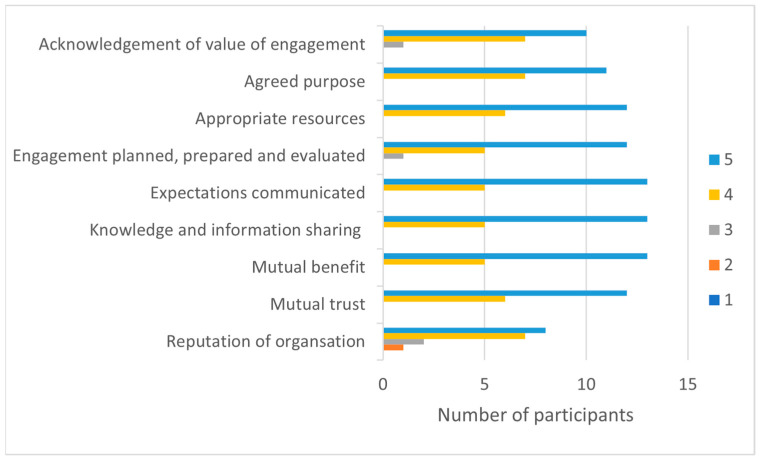
Engagement Survey: Factors facilitating sustained collaboration. Rating: 1 = strongly disagree, 2 = disagree, 3 = neither disagree nor agree, 4 = agree, and 5 = strongly agree.

**Figure 7 ijerph-20-06131-f007:**
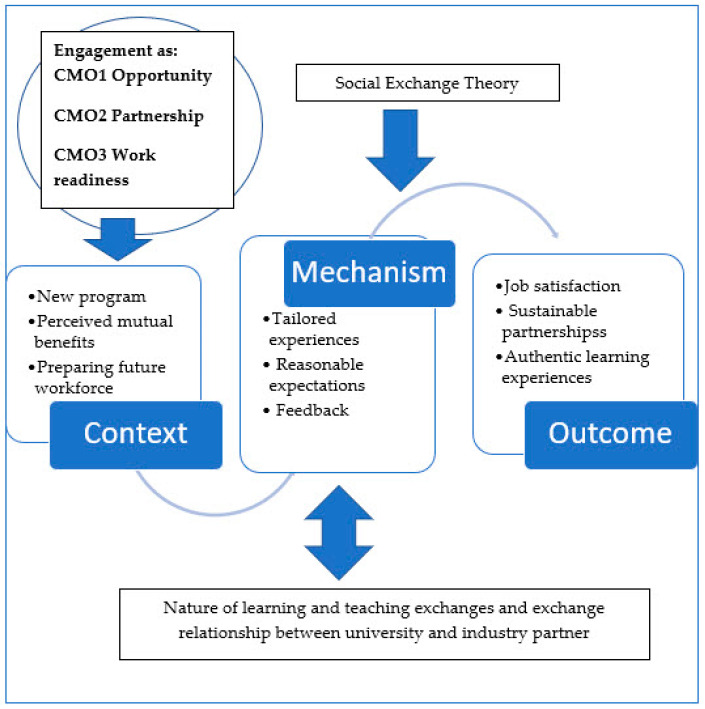
CMO pathways for effective industry engagement in health sciences education.

**Table 1 ijerph-20-06131-t001:** Study-specific Context Mechanism Outcome considerations.

	General Description	Study-Specific Description
**Context**	Individual participant and site factors that may positively or negatively impact engagement experiences.	Large (~9000 km^2^) state-funded health district covering urban and semi-rural areas, servicing ~350,000 people. Diverse workforce, including new graduate and experienced clinicians, providing a range of service delivery models in hospital and community settings. Professional-experience levels of clinicians. COVID-19 leading to increased clinician workload and staff redeployment during study. New degree based in Western Sydney University requiring new curriculum design and development. Aims to create a mutually beneficial, sustainable collaboration between the university and LHD.
**Mechanism**	Collaborative strategies, learning and teaching approaches, and resources implemented during subjects to facilitate engagement and participants’ responses to these approaches.	Curriculum co-design and delivery. Response to resources. Online learning environment. Case-based learning approaches.
**Outcome**	Causal patterns identified when different contexts and/or mechanisms experienced by participants were associated with particular engagement outcomes.	Factors impacting clinicians’ engagement with university learning and teaching. Clinicians’ perceptions of benefits and challenges of engagement. Clinicians’ and academics’ experiences of engagement, including factors that facilitated or were barriers to engagement.

**Table 2 ijerph-20-06131-t002:** Engagement Survey’s demographic data.

	Demographic Characteristics	Participants, N = 18 N (%)
Work Status	Full time Part time	12 (67) 6 (33)
Setting	Community Hospital Missing	7 (39) 8 (44) 3 (17)
Area of Work	Paediatrics Adults Mixture Missing	8 (44) 8 (44) 1 (11) 1 (11)
Years of Experience	18 months to 5 years 5–10 years >10 years	4 (22) 3 (17) 11(61)
Highest Education	Bachelor Master’s PhD or equivalent	13 (77) 4 (22) 1 (11)

## Data Availability

Data is available by contacting the corresponding author.

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
