# Peer review of "Engaging Industry in Health Professions’ Education: Benefits and Challenges"

_ijerph, 2023, doi:10.3390/ijerph20126131_

Round 1
Reviewer 1 Report
I do not find any scientific value in this research. Therefore, I do not find that this research adds to the body of literature on the matter of health education. Figures are poorly presented and the manuscript itself is not easy to follow as it is missing scientific style, flow and clarity. Furthermore, I have specific concerns mentioned below:
line 2 - add study design
line 5 - this is a serious oversight
what is xxx throughout the manuscript?
can you use a more appropriate package for figures?
Author Response
Dear Reviewer,
Thank you for reviewing the manuscript. We were disappointed to hear that you did not find any scientific value in the research or consider that it contributed to the health education literature. The study has adopted a strong theoretical framework and the findings are reported in accordance with realist studies. There is a dearth of literature on industry engagement in health sciences professional preparation programs and we have taken an innovative approach to this gap in health sciences literature. We have made a number of changes in the revised manuscript to enhance structure and clarity, including addition of subheadings, condensing and streamlining content in each section of the manuscript. We respectfully request that you reconsider your decision regarding publication.
Specific reviewer feedback has been addressed as follows:
- Study design has been added to the abstract
- Abstract revised
- The use of xxx was used throughout the manuscript to deidentify authors and sites during the review process. Identifying information has been added to the revised manuscript
- All Figures have been revised
- Manuscript has been carefully edited, condensed and additional headings included in structure
Reviewer 2 Report
The article in general represents the formal result of an investigation, it observes various strengths that deserve its publication. However, there are some structural details that are considered pertinent to be addressed by the authors:
a). Rethink the title in two ways: (1) express in the title the reasons why Engaging is being studied and (2) in a space of few words, the words Engaging and education are repeated. It is necessary to offer a more attractive title that truly explains the purposes of the research.
b). I believe that the manuscript does not observe an adequate architecture in the distribution of the contents, especially in the following sections: (1) the introduction is too long, it should be divided into two: the introduction itself and a theoretical framework; (2) the discussion is too extensive and little classificatory of the results; they should consider making it shorter and more confrontational with the scientific literature, it talks too much about the results of the focus group.
c). It is necessary to make some adjustments in the abstract, especially specifying the participants in the study, I am sorry this aspect is confusing or it is incomplete.
d) In the text it indicates that the "Likert Scale" was used, however, on some occasions it is correct because the scale goes from totally agree to totally disagree, however, a different scale is indicated in some evaluated aspect and there it should say that it is a “Likert Type” scale.
e). Arrange the figures in such a way that the evaluation criteria are presented completely, the texts are cut. In addition, the headings should be removed and these should be considered more as the title of the figure itself.
f). The results presented after each table must be ordered in a classifying manner, using bullets, numbers or subsections with letters. This gives more clarity to the results.
g). Within the subjects participating in the study I did not understand their quantity, both in the response to the electronic survey and in describing the way in which the homogeneity of the participants in the focus group was taken care of and the reasons why it was only a focus group and not two or three as indicated by the methodology itself.
h). Regarding the methodology, mention somewhere what type of study it is: qualitative, quantitative or mixed. It is understood what type it is, but it is never mentioned.
Author Response
Dear Reviewer
Thank you for your comprehensive review and constructive feedback on our manuscript. We have responded to your feedback as follows:
a). Rethink the title in two ways: (1) express in the title the reasons why Engaging is being studied and (2) in a space of few words, the words Engaging and education are repeated. It is necessary to offer a more attractive title that truly explains the purposes of the research.
Thank you for this suggestion. The manuscript title has been revised to "Benefits and Challenges of Engaging Industry in Health Professions Education"
b). I believe that the manuscript does not observe an adequate architecture in the distribution of the contents, especially in the following sections: (1) the introduction is too long, it should be divided into two: the introduction itself and a theoretical framework; (2) the discussion is too extensive and little classificatory of the results; they should consider making it shorter and more confrontational with the scientific literature, it talks too much about the results of the focus group.
1) The introduction has been revised and condensed. The theoretical framework has been summarised and shifted to the methods section.The revisions have significantly decreased the introduction wordcount. Subheadings have been included in the introduction to enhance structure and cohesion. The revisions have significantly decreased the introduction wordcount.
2) The discussion has been revised. Findings from the survey and focus groups have been integrated to reduce duplication, overall word count and to prioritise important findings. The authors have reduced focus-group specific exemplars and added more content that addresses scientific literature, including findings from studies that address issues of job satisfaction and retention in health sciences (please see, for example Ewen et al., 2021 and Heritage et al., 2019).
c). It is necessary to make some adjustments in the abstract, especially specifying the participants in the study, I am sorry this aspect is confusing or it is incomplete.
The abstract has been revised and clearly states the number of survey and focus group participants.
d) In the text it indicates that the "Likert Scale" was used, however, on some occasions it is correct because the scale goes from totally agree to totally disagree, however, a different scale is indicated in some evaluated aspect and there it should say that it is a “Likert Type” scale.
The authors agree that 'Likert- type' scale more appropriately reflects the survey tool and have revised the method section.
e). Arrange the figures in such a way that the evaluation criteria are presented completely, the texts are cut. In addition, the headings should be removed and these should be considered more as the title of the figure itself.
Thank you for these recommendations. The six figures that present findings have been revised accordingly, to ensure evaluation criteria are fully visible, and Figures are relabelled with appropriate titles.
f). The results presented after each table must be ordered in a classifying manner, using bullets, numbers or subsections with letters. This gives more clarity to the results.
The authors have introduced additional subheadings and numbering for each of the survey and focus group findings to improve clarity of the results.
g). Within the subjects participating in the study I did not understand their quantity, both in the response to the electronic survey and in describing the way in which the homogeneity of the participants in the focus group was taken care of and the reasons why it was only a focus group and not two or three as indicated by the methodology itself.
The number of subjects participating in each part of the study (survey 21 respondents, 18 completed entire survey; focus group 5 participants, 2 academics and 3 clinicians). The focus group was a mixed academic/ clinician group and the clinicians were drawn from one large health service but employed at different sites. Hence, we have captured multiperspectives within the focus group. As this is a new and innovative project we perceived that a combined focus group would provide richer findings compared with individual site interviews. However, the authors acknowledge in the limitations of the study that future research is needed to determine if the CMO pathways reflect the experiences of clinicians and academics from other disciplines.
h). Regarding the methodology, mention somewhere what type of study it is: qualitative, quantitative or mixed. It is understood what type it is, but it is never mentioned.
The original manuscript included a statement that the study has adopted a sequential quant/ qual mixed methods approach and this content has now been moved to the top of the Materials and Methods section to clarify the methods for readers.
Reviewer 3 Report
Comments
Line 12: “industry” all together.
Line 16: “obtained” ibidem.
Line 17:” future workforce” ibidem.
Line 22: “accordance” ibidem.
The current environment in the management of most higher education processes requires an approach similar to that shown in the context of any other business. But, in this article in particular, the need for interaction and coordination of activities with the external environment is absolutely imperative.
Modern technology, the greater demands of the market and the desire to combine efficiency with quality assurance are at the root of the emergence of new forms of organization of production. On the approach of the Social Exchange Theory, see, e.g., Simbula et al (2023) “Building Work Engagement in Organizations: A Longitudinal Study Combining Social Exchange and Social Identity Theories”, Michalová et al (2023) “Epistemological Approach to Knowledge Sharing Issues at Universities in the COVID-19 Pandemic: Altruism and Social Exchange Theory Context” and Kaufman et al (2022) “At What Cost? An Evaluation of the Health and Human Services Proposed Rule, "Proposed Modifications to the HIPAA Privacy Rule to Support, and Remove Barriers to, Coordinated Care and Individual Engagement", Political Perspectives, Vol. 29, pp1-13.
Suggestions
I suggest that the rest of the text be subject to revision of the “broken” words as written in Comments, which only takes into account the Abstract.
All scientific research is a human activity of great ethical responsibility due to its inherent characteristics. We highlight the importance of health research, not forgetting the general ethical principles that apply to it. The more this happens, the more research and its positioning as a “human science” of health professionals expands. It is not just clinicians who must have additional concerns for the well-being of study subjects and respect for the rights and integrity of individuals.
My suggestion is that the article includes an ethical reference to the approach to the subject of the subject studied. The look of ethics in research covers all stages of the research process, as a concern with the ethical quality of procedures and with respect for established principles and not just the “engagement with health professions curriculum” (Line 63). See, e.g., Amber et all (2022) “Ethical Case Studies for Advanced Practice Nurses: Solving Dilemmas in Everyday Practice”.
Another suggestion for future investigations is to use another type of software stronger and more powerful that does not IBM SPSS.
Author Response
Dear Reviewer,
Thank you for your review and helpful suggestions. We have responded to your feedback as follows:
The current environment in the management of most higher education processes requires an approach similar to that shown in the context of any other business. But, in this article in particular, the need for interaction and coordination of activities with the external environment is absolutely imperative.
Modern technology, the greater demands of the market and the desire to combine efficiency with quality assurance are at the root of the emergence of new forms of organization of production. On the approach of the Social Exchange Theory, see, e.g., Simbula et al (2023) “Building Work Engagement in Organizations: A Longitudinal Study Combining Social Exchange and Social Identity Theories”, Michalová et al (2023) “Epistemological Approach to Knowledge Sharing Issues at Universities in the COVID-19 Pandemic: Altruism and Social Exchange Theory Context” and Kaufman et al (2022) “At What Cost? An Evaluation of the Health and Human Services Proposed Rule, "Proposed Modifications to the HIPAA Privacy Rule to Support, and Remove Barriers to, Coordinated Care and Individual Engagement", Political Perspectives, Vol. 29, pp1-13.
The authors agree that contemporary issues of modern technology, market demands of the market and needs to balance efficiency with quality are highly relevant to the tertiary education sector. Thank you for recommending additional references. Simbula et al (2023) and Michalová et al (2023) have been incorporated within the introduction section of the manuscript. The authors also read Kaufman's (2022) work with interest. Kaufman’s article provides a strong case for the need for external stakeholder input but given the specific context of this study and other references providing similar content, it was not included.
I suggest that the rest of the text be subject to revision of the “broken” words as written in Comments, which only takes into account the Abstract.
Thank you. The authors agree that 'broken words' detract from the cohesiveness of the manuscript. This was not a feature of the original manuscript and appears to be a function of journal formatting. However, the authors have revised sentence structuring to avoid this occurring whenever possible. This issue may also be further addressed during editing for publication.
All scientific research is a human activity of great ethical responsibility due to its inherent characteristics. We highlight the importance of health research, not forgetting the general ethical principles that apply to it. The more this happens, the more research and its positioning as a “human science” of health professionals expands. It is not just clinicians who must have additional concerns for the well-being of study subjects and respect for the rights and integrity of individuals.
My suggestion is that the article includes an ethical reference to the approach to the subject of the subject studied. The look of ethics in research covers all stages of the research process, as a concern with the ethical quality of procedures and with respect for established principles and not just the “engagement with health professions curriculum” (Line 63). See, e.g., Amber et all (2022) “Ethical Case Studies for Advanced Practice Nurses: Solving Dilemmas in Everyday Practice”.
The authors agree with the reviewer's statements regarding the critical importance of ensuring that all stages of the research project are ethically sound and that partnerships between universities and academics uphold principles of ethical practice. We have added a statement to the content provided on HREC ethics approval to note that the study has followed the guidelines and values espoused by the Australian Research Council and the Code for the Responsible Code of Research that binds all Australian researchers.
Another suggestion for future investigations is to use another type of software stronger and more powerful that does not IBM SPSS.
Thank you for this suggestion. We appreciate that there are other statistical analysis software currently available such as R or STATA. In this instance, we feel that the IBM SPSS software is sufficient to complete the needed analysis for the study but future projects may benefit from alternative analysis programs.
Reviewer 4 Report
The manuscript reports about a mixed-method study according to a realist approach. The topic is the engagement between the university and non-academic professionals in health professions education. The topic is very relevant and - to my surprise - the statement of the Authors that "Surprisingly, there is limited evidence to guide academic-industry collaboration" (page 1, lines 38-39) is true! I checked ...
I have few suggestions, that has to do with the readibility of the article:
1- the term "industry": maybe that in some contexts the term has a wider meaning and use than in everyday use (the companies and activities involved in the process of producing goods for sale, especially in a factory or special area - Cambridge dictionary) , but I suggest that the term should be defined, as for example in ref #18)
2 - The Introduction is rather long and the reader would benefit of some subheadings, for example according to the well know structure Problem-Gap-Hook (https://www.ncbi.nlm.nih.gov/pmc/articles/PMC4602011/).
I found confusing the sudden appearance of the "Engaging Industry for Engaging Education project". Maybe this reference could go in the Discussion. I also wonder if the description of the SET models could go in the Method section.
3 - Results: in many realist articles, the final theory is graphically represented. I suggest that the authors could think of a graphic model, helping the reader to get the sense of the whole in browsing the long Table 3.
Overall, with these minor revisions, the article is can be accepted
Author Response
Dear Reviewer,
Thank you for constructive and helpful review. We have responded to your feedback as follows:
The manuscript reports about a mixed-method study according to a realist approach. The topic is the engagement between the university and non-academic professionals in health professions education. The topic is very relevant and - to my surprise - the statement of the Authors that "Surprisingly, there is limited evidence to guide academic-industry collaboration" (page 1, lines 38-39) is true! I checked ...
Thank you. We are pleased that you have also noted that this is a topic that requires further empirical attention.
1- the term "industry": maybe that in some contexts the term has a wider meaning and use than in everyday use (the companies and activities involved in the process of producing goods for sale, especially in a factory or special area - Cambridge dictionary) , but I suggest that the term should be defined, as for example in ref #18)
The authors have adopted Manwaring et al.'s definition, as stated in the first paragraph of the introduction, to clarify the term 'industry' for readers.
2.
2 - The Introduction is rather long and the reader would benefit of some subheadings, for example according to the well know structure Problem-Gap-Hook (https://www.ncbi.nlm.nih.gov/pmc/articles/PMC4602011/).
I found confusing the sudden appearance of the "Engaging Industry for Engaging Education project". Maybe this reference could go in the Discussion. I also wonder if the description of the SET models could go in the Method section.
Thank you for these suggestions. The authors have restructured the introduction section of the manuscript. The content has been condensed and discussion of the theoretical framework has been shifted to the Method section. The authors have included subheadings to align with the problem-gap- hook approach and introduced the Engagement Project within this structure to clarify how this project aligns with the research questions.
3 - Results: in many realist articles, the final theory is graphically represented. I suggest that the authors could think of a graphic model, helping the reader to get the sense of the whole in browsing the long Table 3.
The authors agree that a graphic model may provide a clearer, more holistic presentation of key study findings, in keeping with realist projects. The authors have replaced Table 3 with Figure 7.
Round 2
Reviewer 2 Report
It is impossible to appreciate a document with all the changes made without accepting them (the author should have sent a clean article). I think my observations were heeded.